# Potential of an Interorgan Network Mediated by Toxic Advanced Glycation End-Products in a Rat Model

**DOI:** 10.3390/nu13010080

**Published:** 2020-12-29

**Authors:** Shinya Inoue, Takanobu Takata, Yusuke Nakazawa, Yuka Nakamura, Xin Guo, Sohsuke Yamada, Yasuhito Ishigaki, Masayoshi Takeuchi, Katsuhito Miyazawa

**Affiliations:** 1Department of Urology, Kanazawa Medical University, Uchinada, Ishikawa 920-0293, Japan; nkzw-y@kanazawa-med.ac.jp (Y.N.); miyazawa@kanazawa-med.ac.jp (K.M.); 2Department of Advanced Medicine, Medical Research Institute, Kanazawa Medical University, Uchinada, Ishikawa 920-0293, Japan; takajjjj@kanazawa-med.ac.jp (T.T.); takeuchi@kanazawa-med.ac.jp (M.T.); 3Division of Molecular and Cell Biology, Department of Life Sciences, Medical Research Institute, Kanazawa Medical University, Uchinada, Ishikawa 920-0293, Japan; yuka-n@kanazawa-med.ac.jp (Y.N.); ishigaki@kanazawa-med.ac.jp (Y.I.); 4Department of Pathology and Laboratory Medicine, Kanazawa Medical University, Uchinada, Ishikawa 920-0293, Japan; xinguo@kanazawa-med.ac.jp (X.G.); sohsuke@kanazawa-med.ac.jp (S.Y.)

**Keywords:** toxic advanced glycation end-products (TAGE), high-fructose corn syrup (HFCS), serum levels of TAGE, intracellular TAGE, lifestyle-related diseases (LSRD), kidney, microarray

## Abstract

Excessive intake of glucose and fructose in beverages and foods containing high-fructose corn syrup (HFCS) plays a significant role in the progression of lifestyle-related diseases (LSRD). Glyceraldehyde-derived advanced glycation end-products (AGEs), which have been designated as toxic AGEs (TAGE), are involved in LSRD progression. Understanding of the mechanisms underlying the effects of TAGE on gene expression in the kidneys remains limited. In this study, DNA microarray analysis and quantitative real-time polymerase chain reaction (PCR) were used to investigate whether HFCS-consuming Wister rats generated increased intracellular serum TAGE levels, as well as the potential role of TAGE in liver and kidney dysfunction. HFCS consumption resulted in significant accumulation of TAGE in the serum and liver of rats, and induced changes in gene expression in the kidneys without TAGE accumulation or upregulation of receptor for AGEs (RAGE) upregulation. Changes in specific gene expression profiles in the kidney were more correlated with TAGE levels in the liver tissue than in the serum. These findings suggest a direct or indirect interaction may be present between the liver and kidneys that does not involve serum TAGE or RAGE. The involvement of internal signal transduction factors such as exosomes or cytokines without IL-1β and TNF-α is suggested to contribute to the observed changes in kidney gene expression.

## 1. Introduction

Increasing evidence suggests that excessive sugar intake leads to the development of a variety of lifestyle-related diseases (LSRD) [1,2]. The relationship between obesity and the intake of soft drinks with added sugars and obesity has been well established, particularly in developed countries [3,4,5,6]. The World Health Organization has recently recommended that the daily intake of added sugars should not exceed 5% of the total energy intake [7]. High-fructose corn syrup (HFCS), a liquid sweetener with fructose and glucose, is present in most soft drinks at an estimated concentration of 10% [8]. Although fructose and glucose are necessary carbohydrates that provide energy to the human body [9], most of the ingested HFCS uses adenosine triphosphate when metabolized by the liver, resulting in the development of gout, hypertension, cardiovascular disease (CVD), and renal dysfunction [10]. However, the mechanisms of the renal dysfunction caused by HFCS intake cannot be evaluated solely based on uric acid levels. This study focused on the effects of advanced glycation end-products (AGEs) on the kidneys in a rat model. Although a variety of AGEs are formed in the body [11], in response to excessive intake of foods and drinks containing sugars, we have previously designated glyceraldehyde (GA; an intermediate of glucose/fructose metabolism)-derived AGEs as toxic AGEs (TAGE) because of their cytotoxicity and involvement in LSRD [12,13]. It has been shown that TAGE in the liver of patients with non-alcoholic steatohepatitis (NASH) are higher than those in the liver of patients with simple steatosis [14,15]. Intracellular TAGE are produced in hepatocytes [16,17], pancreatic cells [18], neuroblasts [19], and cardiomyocytes [20] and induce cell death in vitro [21]. Similarly, serum levels of TAGE increased in patients with LSRD, including NASH; arteriosclerosis; diabetes; CVD; and infertility [13]. It has also been demonstrated that TAGE present in the blood induce cytotoxicity in cells via receptor for AGEs (RAGE) [13,15]. As the liver is the main organ involved in sugar metabolism, it is speculated that large amounts of intracellular TAGE are generated in the liver, then transported to the blood, and induces responses in other organs [13,15]. However, no studies have examined the effects of TAGE generation on renal tissues. GA is generated in the liver via three pathways [15,22]. As the kidneys express the enzymes sorbitol dehydrogenase [23], aldose reductase [24], fructokinase [25], and aldolase B [26], we hypothesized that the three pathways involved in GA generation existed in the kidneys, similar to the liver. Therefore, we hypothesize that TAGE are generated in the kidneys. We aimed to understand the effects of TAGE generation on liver and kidneys in vivo by investigating its effects on gene expression in renal tissues. Furthermore, as RAGE is expressed in the kidneys, we considered the possibility that TAGE present in the blood induced renal dysfunction via RAGE [27]. We focused on the possibility that liver generates intracellular TAGE, which in turn influence the kidneys. Results obtained indicated that direct or indirect interaction between the liver and kidneys does not involve serum TAGE or RAGE, and the involvement of internal signal transduction factors such as exosomes or cytokines are suggested as potential factors contributing to the observed alterations in kidney gene expression. Additional preclinical and clinical studies are, nevertheless, needed to determine the relevance of targeting TAGE and RAGE to reduce the complications of LSRD, and to identify new therapeutic interventions that may reduce or delay obesity and mortality in the human population.

## 2. Materials and Methods 

### 2.1. Animal Experiments

The study was approved by the Institutional Animal Care and Use Committee of Kanazawa Medical University, and experiments were conducted in accordance with the Institutional Animal Care and Use Guidelines following the “Basic Guidelines for the Conduct of Animal Experiments” and “Basic Guidelines for Animal Care Management” (license number: 2017-59).

After 1 week of acclimation to the rearing environment, 11-week-old male Wistar rats (SLC, Shizuoka, Japan; *n* = 20) were divided into two groups (control and HFCS, 10 animals each), which were matched for body weight. The animals were housed individually and were provided a standard feed (Labo MR stock, Nosan, Kanagawa, Japan) and commercially available water (Evian, Évian-les-Bains, France). In the HFCS group, commercial sweeteners were added to the drinking water at a total concentration of 137 g/L HFCS55 (fructose, 41.25 g; glucose, 33.75 g; water, 25.0 g for a total of 100 g; 1.35 g/mL sugar; 276 kcal/100 g) to prepare a 10% HFCS solution. Water consumption was measured daily, and the body weight was measured weekly. The animals were 12 weeks of age at the start of the study, reared for 8 weeks, and euthanized at 20 weeks. The animals were moved to a metabolic cage 24 h before euthanasia, and urine was collected under fasting and ad libitum drinking conditions. After the rats were anaesthetized by isoflurane inhalation, blood was collected from the inferior vena cava via a midline incision in the abdomen, and the kidneys and liver were removed.

Blood tests, including blood urea nitrogen (BUN), creatinine, uric acid, low-density lipoprotein cholesterol (LDL-C), triglycerides, glucose, and glycated hemoglobin (HbA1c), were performed by Oriental Yeast Co., Ltd. (Tokyo, Japan) [28]. Urinalysis was performed by SRL. (Tokyo, Japan) [29] was conducted to determine the urine volume, pH, and the concentrations of calcium, creatinine, uric acid, and 8-hydroxy-2′-deoxyguanosine (8-OhdG). To measure oxalic and citric acids, the pH was adjusted to ≤2 by adding hydrochloric acid (Wako Pure Chemical Co., Ltd. Osaka, Japan). Chemiluminescence analysis was conducted using a ruthenium complex and a FOM-110 A analytical instrument (Hokuto Denko Co, Ltd., Tokyo, Japan) [30].

Kidney and liver tissue samples were fixed in 10% formalin, embedded in paraffin, and cut into 3 μm sections. After hematoxylin-eosin staining, the samples were evaluated histologically using an Olympus BX50 light microscope (Olympus, Tokyo, Japan).

### 2.2. Measurement of TAGE in the Serum and Hepatic and Renal Tissues

Serum levels of TAGE were measured using a competitive ELISA and an immunopurified TAGE antibody [31]. Briefly, 96-well microtiter plates were coated with the TAGE antibody and incubated with 1 μg/mL TAGE-bovine serum albumin (BSA) (TAGE-BSA) overnight at 4 °C. The wells were washed thrice with 0.3 mL phosphate-buffered saline-Tween 20 (PBST) and then were blocked with 0.2 mL of PBS containing 1% BSA by incubation for 1 h. After washing the wells with PBST, the test samples (50 μL) were added to each well as a competitor for 50 μL of TAGE antibody (1:1000), followed by incubation for 2 h at room temperature with gentle shaking on a horizontal rotary shaker. The wells were washed with PBST and developed with an alkaline phosphatase-linked anti-rabbit IgG using p-nitrophenyl phosphate as a colorimetric substrate. The results are expressed as TAGE units (U) per mL of serum, with 1 U corresponding to 1 μg of TAGE-BSA standard, as described previously [32]. The sensitivity of the assay and intra-assay coefficients of variation were 0.01 U/mL, 6.2%, and 8.8%, respectively [32].

For a slot blot analysis of liver TAGE, the following materials were purchased: 3-[(3-cholamidopropyl)dimethylammonio]-1-propane sulfate (CHAPS), Dojindo Laboratories (Kumamoto, Japan); ethylenediamine-N, N, N, N’-tetraacetic acid (EDTA)-free protease inhibitor cocktail, Roche Applied Science (Penzberg, Germany); Bradford method protein assay kit, Takara Bio (Otsu, Japan); horseradish peroxidase (HRP)-linked molecular marker, Bionexus (Oakland, CA, USA); and HRP-linked goat anti-rabbit IgG antibody, Dako (Glostrup, Denmark). All other reagents and kits were purchased from Wako Pure Chemical Industries (Osaka, Japan). Tissue blocks were washed with ice-cold PBS(−) and immersed in liquid nitrogen for 30 s. Blocks were placed in tubes to which the buffer [a solution of 2 M thiourea, 7 M urea, 4% CHAPS, and 30 mM Tris (hydroxymethyl) aminoethane], and a buffer [radioimmunoprecipitation assay buffer (Thermo Fisher Scientific Inc., Waltham, MA, USA) and solution of the EDTA-free protease inhibitor cocktail (9:1)]. The measure of TAGE in the tissue was performed as described [18,20,33]. TAGE-BSA and an anti-TAGE antibody were prepared as described previously [32]. The protein assay was performed as previously described with some modifications [33]. Cell lysates (2.0–30.0 μg of protein/lane) were blotted onto a polyvinylidene difluoride (PVDF) membrane.

### 2.3. Total RNA Extraction

RNA was extracted from homogenized on kidney samples using the miRNeasy mini kit (Qiagen, Hilden, Germany) according to the manufacturer’s instructions. The extracted RNA was measured at 260/280 nm using a NanoDrop spectrophotometer (Thermo Fisher Scientific, Waltham, MA, USA). RNA quality was determined using the RNA 6000 Nano Kit (Agilent Technologies, Santa Clara, CA, USA). Samples with the RNA integrity number of more than 6 were used as templates for cDNA synthesis.

### 2.4. Transcriptome Analysis by DNA Microarray

The RNA extracted from the control (*n* = 3) and HFCS (*n* = 3) groups was analyzed using the Rat Gene ST 2.0 microarray (Affymetrix, High Wycombe, UK). The GeneSpring GX software package version 14.9 (Agilent) was used to generate a volcano plot of differentially expressed genes with a difference >1.5-fold, and genes whose mRNA expression levels differed significantly (*p* < 0.05) between the HFCS and control groups were identified using a *t*-test. No multiple testing correction was performed. Intermolecular pathway analysis of the obtained DNA microarray data was performed using Ingenuity pathway analysis (Ingenuity Systems, Redwood City, CA, USA).

### 2.5. Gene Expression Analysis by Quantitative Real-Time Polymerase Chain Reaction (qPCR)

cDNA was synthesized from RNA samples using the SuperScript IV VILO master mix with ezDNase (Invitrogen, Carlsbad, CA, USA). qPCR was performed using the cDNA and TaqMan gene expression assays with the TaqPath qPCR master mix, CG (Applied Biosystems, Foster City, CA, USA). The following TaqMan assays were used: *Cyp24a1*, Rn01423143_m1; *Plin2*, Rn01399516_m1; *Usp2*, Rn01399516_m1; *Calb1*, Rn00583140_m1; *Ager*, Rn01525753_g1; β-actin, Rn00667869_m1. Expression was determined using a QuantStudio 12K Flex real-time PCR system (Applied Biosystems, Foster City, CA, USA). The relative expression of each target gene was normalized to that of β-actin.

### 2.6. Total Protein Extraction of Renal Tissue for Western Blot (WB) Analysis

Radioimmunoprecipitation assay (RIPA) buffer and BCA method protein assay kit were obtained from Thermo Fisher Scientific Inc. Tissue blocks were washed with ice-cold PBS(−) and immersed in liquid nitrogen for 30 s. Blocks were placed in tubes to a buffer [RIPA buffer and solution of the EDTA-free protease inhibitor cocktail (9:1)]. Methods for the homogenization and collection of tissue lysates were described in the method for the SB analysis. 

### 2.7. Western Blot Analysis of Expression of RAGE on the Renal Tissue

Anti-GAPDH antibody was obtained from abcam (Tokyo, Japan) and HRP-linked goat anti-mouse IgG antibody was purchased from Thermo Fisher Scientific Inc. Protein concentrations were assessed using the protein assay kit for the BCA method with BSA as a standard. Tissue lysates (30 μg of protein) were mixed with sodium dodecyl sulfate (SDS) sample buffer and 2-mercaptoethanol (Sigma-Aldrich, St. Louis, MO, USA), and then heated at 95 °C for 5 min. Equal amounts of cell extracts were resolved by 4–15% gradient SDS-polyacrylamide gel (Bio-Rad, CA, USA) electrophoresis and transferred onto polyvinylidene difluoride (PVDF) membrane (0.45 μm; Millipore, MA, USA). Membranes were blocked at r.t. for 30 min using 5% skimmed milk in PBS-T. We then used PBS-T for washing or as the solvent of antibodies. After washing twice, membranes were incubated with the anti-RAGE antibody (1:1000; ab3611) at 95 °C for overnight. PVDF membrane was washed four times with PBS-T and incubated with HRP-linked donkey anti-rabbit IgG antibody (1:5000; Prodcut No. P0448, DAKO) at r.t. for 1 h. Membrane was then washed with PBS-T. Immunoreactive complexes were visualized using the ImmunoStar LD kit. Band densities on the membranes were measured using Fusion FX fluorescence imager (M&S Instruments Inc., Osaka, Japan) and expressed in arbitrary units (AU). Equivalent sample loading was confirmed by stripping membranes with the Western stripping solution (Nacalai Tesque), and this was followed by blotting with the anti-GAPDH antibody (1:40,000; ab8245) and HRP-linked goat anti-mouse IgG antibody (1:10,000; Product No. 31432). 

### 2.8. Oil Red O Staining of the Kidney and the Liver

Liver and kidney specimens from the control and HFCS groups were frozen. Oil Red O was dissolved at a concentration of 6.1 mM in isopropanol and diluted with water to a concentration of 3.7 mM. Tissue sections were washed with water and then incubated with the Oil Red O solution at room temperature for 10 min, followed by the removal of Oil Red O with water, incubation with hematoxylin at room temperature for 5 min, and washing with water. Finally, the sections were coated with malinol mounting medium and observed under a NanoZoomer slide scanner (Hamamatsu Photonics K.K., Hamamatsu, Japan).

### 2.9. Measurement of Cytokines in the Serum

Cytokine levels were measured in the serum obtained from rats in the control and HFCS groups. The levels of interleukin (IL)-1β were measured using a rat IL-1β ELISA kit (#27193; IBL, Gunma, Japan), and those of tumor necrosis factor (TNF)-α were measured using a rat TNF-α Quantikine ELISA kit (RTA00; R&D Systems, Minneapolis, MN, USA) according to the manufacturers’ instructions.

### 2.10. Immunostaining of CD68 in the Kidney

To evaluate the infiltration of macrophages into kidney tissue, we performed immunohistochemistry (IHC) using a rabbit anti-rat CD68 antibody (1:500, ab125212; Abcam plc, Cambridge, UK) and paraffin sections. After deparaffinization and rehydration, the specimens were incubated with 1% hydrogen peroxide for 15 min at room temperature, then heated at 95 °C for 20 min for antigen retrieval and probed with the primary antibody for 1 h at room temperature. IHC images were captured using a DP26 digital camera (Olympus, Tokyo, Japan).

### 2.11. Statistical Analysis

Statistical analysis was performed using the EZR software (Saitama Medical Center, Jichi Medical University, Saitama, Japan) [34] based on R (The R Foundation for Statistical Computing, Vienna, Austria). The Mann-Whitney *U*-test or Student’s *t*-test was used to compare mean values between the two groups. A value of *p* < 0.05 was considered significant.

The correlations between the TAGE levels in the serum or liver tissue and gene expression levels measured by qPCR were evaluated using Pearson’s integral correlation coefficient. A value of *p* < 0.05 was considered significant.

## 3. Results

### 3.1. HFCS Loading Altered Triglyceride and HbA1c Levels in Wistar Rats

No significant differences in the body weights were observed between the control and HFCS groups (Appendix A). However, water intake and urine excretion were higher in the HFCS group than those in the control group. Urinalysis revealed a lower urine pH in the HFCS group than that in the control group. No significant differences were observed in the concentrations of creatine, uric acid, calcium, oxalic acid, citric acid, and 8-OHdG between the two groups (Appendix A). Blood tests showed elevated concentrations of BUN, triglycerides, and HbA1c in the HFCS group compared with those in the control group. No significant differences were observed in the levels of creatinine, uric acid, LDL-C, or blood glucose between the two groups (Appendix A). Thus, significant differences (*p* < 0.05) between the HFCS and control groups were only observed in urinary pH and serum triglyceride and HbA1c levels.

### 3.2. Histological and IHC Evaluation of Renal and Hepatic Tissues

Histological evaluation of liver and kidney tissues using hematoxylin and eosin staining revealed no morphological differences between the control and HFCS groups (Figure 1). Oil Red O staining of the kidney and liver tissues revealed no areas of positive staining in both groups (Appendix A). CD68 immunostaining of kidney tissue did not reveal the presence of positive cells in the control and HFCS groups (Appendix A).

### 3.3. CD68 Imunnostaining of Renal Tissue

CD68 immunostaining of the kidneys showed no positive cells in the control and HFCS groups (Appendix A).

### 3.4. HFCS Caused Elevation of TAGE Levels in the Serum and Liver but Not in the Kidneys

The mean serum TAGE level was 1.14-fold higher in the HFCS group than that in the control group (16.21 ± 1.77 versus 14.27 ± 1.07 U/mL, respectively), and the mean TAGE level in the liver was 1.74-fold higher in the HFCS group than that in the control group (1.53 ± 0.41 versus 0.88 ± 0.18 µg/mg protein, respectively; Figure 2). However, the band intensity of intracellular TAGE from renal tissue lysates did not increase with the increase in the total protein mass in the slot blot analysis (Appendix A), and therefore, no quantification was performed.

### 3.5. HFCS Did Not Affect the RAGE Gene and Protein Expression in the Kidneys

No significant difference in the expression levels of the RAGE-encoding gene in the kidneys was observed between the HFCS and control groups of rats using qPCR (Appendix A). Western blot analysis of the expression of the RAGE protein in the kidneys did not show any significant difference between the two groups of rats either (Appendix A).

### 3.6. HFCS Caused Changes in the Expression of 12 Annotated Kidney Genes

The effect of HFCS on gene expression in the kidneys was investigated using DNA microarray analysis. In the three samples from each group, a significant differences (>1.5-fold) were observed in 28 genes (*p* < 0.05), 12 of which were well-annotated. We observed that HFCS loading resulted in increased expression of eight annotated genes (*Hbb*, *Hba1* or *Hba2*, *Cyp24a1*, *Bcl6*, *Plin2*, *Wsb1*, *Mlph*, and *Fermt1;*
Table 1) and decreased expression of four annotated genes (*Fkbp5*, *Calb1*, *Scnn1a*, and *Usp2*; Table 2). Ingenuity pathway analysis of an intermolecular network revealed that nearly all the genes with altered expression were involved in fructose metabolism (except for *Hba1/Hba2*, *Wsb1*, and *Mlph*; Figure 3), thus suggesting a novel relationship between the latter three genes and fructose uptake.

DNA microarray analysis was used to investigate the kidney expression of four genes (*Cyp24a1*, *Plin2*, *Calb1 and Usp2*), which are reported to be involved in metabolic syndrome and LSRD. The analysis of ten samples from each group revealed significant intergroup differences in the expression of *Plin2*, *Calb1*, and *Usp2*, but not *Cyp24a1*. These results were confirmed by real-time PCR (Figure 4).

### 3.7. Altered Expression of Usp2 and Calb1 Significantly Correlated with TAGE Generation in the Liver

As no accumulation of TAGE was observed in the kidneys, the observed changes in gene expression in the renal tissue could be attributed to other consequences of HFCS loading. To determine whether these changes were caused by serum or liver TAGE, we investigated the relationships between serum and liver TAGE levels and the mRNA expression levels of the four genes determined using qPCR. The data revealed that the serum TAGE concentrations did not correlate with the expression levels of *Plin2*, *Cyp24a1*, *Usp2*, or *Calb1* in the kidney (Figure 5). By contrast, the expression of *Usp2* and *Calb1* negatively correlated (*p* < 0.001) with TAGE concentrations in liver tissue (Figure 6). This significant correlation did not seem to depend on the HFCS load, as the expression of neither *Plin2* nor *Cyp24a1* was affected. Thus, TAGE generation in liver tissue, but not the serum concentration of TAGE, affected the expression of *Usp2* and *Calb1* in the kidneys.

### 3.8. Comparison of Serum IL-1β and TNF-α Levels between the HFCS and Control Groups

To test the hypothesis that some cytokines are involved in gene expression in the kidney, we measured the serum concentrations of IL-1β and TNF-α. IL-1β tended to increase in the HFCS group, but no significant differences were observed in the levels of IL-1β between the two groups (Appendix A). Meanwhile, TNF-α was not detected in either group.

## 4. Discussion

In this study, we investigated the potential effects of HFCS-induced TAGE on the kidneys in a rat model. In vivo experiments indicated that water intake and the volume of urine were higher in the HFCS group than in the control group. However, the body weights did not differ between the groups (Appendix A), despite the higher carbohydrate consumption in the HFCS group. A previous study reported that consumption of 10% HFCS for 13 weeks did not affect the body weights of Wistar/ST rats [33], likely due to a decrease in food consumption by rats receiving HFCS in water. Urinalysis revealed no significant differences in creatine, uric acid, calcium, oxalate, citrate, and 8-OHdG concentrations between the HFCS and control groups (Appendix A). Blood chemistry tests revealed no significant differences in creatine, uric acid, glucose, and LDL-C concentrations between the two groups (Appendix A). Sugar loading increases urinary calcium excretion [35], decreases citrate excretion, and increases oxidative stress [36], as observed in diabetes and renal function impairment. Although excessive consumption of sugar, especially HFCS, is thought to be a cause of obesity [37,38], the conditions employed in our study might not been adequate to induce obesity. We thus concluded that renal dysfunction was not confirmed by renal and hepatic tissue analyses or blood sampling data (Appendix A, Figure 1, and Appendix A). Since even simple steatosis was not shown in the hepatic tissue, we considered that NAFLD did not occur (Figure 1 and Appendix A).

The majority of HFCS components (glucose and fructose) are metabolized in the liver to produce GA; therefore, TAGE are thought to be mainly produced in the liver [13,15]. In the present study, HFCS ingestion by Wistar rats for 8 weeks resulted in a TAGE level of 1.53 μg/mg protein in the liver and 16.21 U/mL in the serum, suggesting that HFCS increased the level of TAGE in the liver, from which they were released them into the blood (Figure 2). Though NAFLD did not occur, we considered that the liver which generated TAGE should be induced the cytotoxic event because intracellular TAGE induce cytotoxicity against hepatic cells [16,17]. In vitro, high concentration of GA generated TAGE in the hepatic cells, and they induce cell death. We considered that intracellular TAGE and other component (e.g., proteins in the cells) would be secreted/released. On the other hand, TAGE in the kidneys were not able to analyze with our slot blot method (Appendix A). A small quantity of TAGE, too low to be detected in this study, may have been produced in the kidneys. Therefore, effects of intracellular TAGE in the kidney was not analyzed. However, the amount of intracellular TAGE which can induce cell death was more than 6.0 μg/mg protein in the hepatic cells, pancreatic cells, cardiomyocytes, and myoblasts cells [17,18,20,21]. Since if intracellular TAGE were generated in the kidney, they might not induce cell death of dramatically dysfunction.

Then, we focused the blood TAGE which may be secreted/released from organs which generated intracellular TAGE [15,21]. One study reported that TAGE in the blood promoted the apoptosis of human renal mesangial cells in vitro [11,39]. Based on our findings and these reports, we hypothesize that TAGE in the blood induce renal dysfunction via RAGE. HFCS administration induced the upregulation of serum levels of TAGE and did not induced gene and protein expression of RAGE (Figure 2A, and Appendix A). We considered the possibility that TAGE in the blood would induce this function of kidney before we analyzed genes expression of kidney and correlation of them (Table 1 and Table 2 and Figure 3, Figure 4 and Figure 5). However, we observed no correlation between serum TAGE and gene expression of the kidney (Figure 5). 

HFCS loading resulted in altered expression of four genes in the kidneys, which was confirmed by DNA microarray (Table 1 and Table 2) and qPCR analyses (Figure 4). Upregulation of *Cyp24a1* and downregulation of *Calb1* were observed in the HFCS group relative to their expression in the control group. CYP24A1 inactivates bioactive vitamin D [40,41], and CALB1 induces the expression of a calcium transporter [42]. Previous studies have reported renal damage in rats fed a high-fructose diet [40,41,42,43,44]. Douard et al. [40,41] reported that a high-fructose diet decreased the expression of calcium transporters such as CALB1 in the kidney and the activity of 1α-hydroxylase, which is involved in the activation of vitamin D in proximal tubules of the kidney. These changes in gene expression result in increased urinary calcium excretion. We observed higher urinary calcium excretion in the HFCS group than that in the control (Table 1).

Upregulation of *Plin2* and downregulation of *Usp2* were observed in the HFCS group. The perilipin gene encodes a protein that is associated with the surface of lipid storage droplets inside cells. Known perilipin family members include PLIN1 [45], PLIN2 (also known as adipophilin, adipose differentiation-related protein, or ADRP) [46], PLIN3 (tail-interacting protein 47, TIP 47) [47], PLIN4 (S3-12) [48,49], and PLIN5 (PAT1, LSPD5, OXPAT, or MLDP) [50,51]. PLIN2, which was elevated with HFCS administration in this study, is expressed systemically around small intracellular lipid droplets and is thought to be responsible for intracellular lipid transfer. Takahashi et al. reported that lipid droplet formation in adipocytes promotes fat synthesis via activation of SREBP-1 and that promotes further lipid droplet formation [52]. Decreased expression of USP2 induced by HFCS administration is reported to contribute to inflammation in adipose tissue and increase low-density lipoprotein (LDL) cholesterol [53,54]. Knockdown of *Usp2* expression in the human myeloid cell line HL-60 is reported to alter the expression of genes involved in the progression of chronic inflammation and type 2 diabetes in adipose tissues, including *aP2*, *PAI-1*, and *MCP-1* [53]. Nelson et al. also reported that the inducible degrader of the LDL receptor (LDLR; IDOL), which promotes dissimilation of the LDLR and is a regulator of serum LDL cholesterol, is regulated by USP2 [54]. Deji et al. reported that feeding of high-fat diet to C57BL/6 mice resulted in renal lipid accumulation and histological changes in the kidneys [55]. In our study, no accumulation of fat droplets was observed in the kidneys. However, induction of expression of genes that induce fat accumulation and genes that contribute to lipid elevation such as LDL cholesterol by HFCS intake may lead to local metabolic changes in the kidney and future renal dysfunction.

Notably, the kidney genes affected by HFCS consumption, *Usp2* and *Calb1*, showed a significantly negative correlation with TAGE concentration in liver tissue (Figure 6) but not in the serum (Figure 5). Thus, the effect of liver-generated TAGE on gene expression in the kidneys does not involve its secretion into the blood to cause changes in kidney gene expression but rather involves factors that mediate between organs. Though we could not prove the existence of factors which are secreted/released from the liver and affected kidney gene expression, they might be exosomes or cytokines. Previous studies reported the secretion of exosomes or cytokines from hepatocytes and liver-associated immune cells [56,57]. The kidneys are reported to be affected by exosomes [58]. It has also been reported that adiponectin and fetuin-A may act as mediators between fat and the kidney and liver organs [59]. However, IL 1-β and TNF-α was not increased by HFCS administration (Appendix A). If cytokines would be the factor which induce downregulation of *Usp2* and *Calb1*, IL-1βand TNF-α might not be these factors. Elucidation of TAGE metabolism in vivo and the molecular mechanism underlying the effects reported herein require further investigation.

## 5. Conclusions

This study showed that HFCS ingestion altered the expression of *Cyp24a1* and *Calb1*, which encode inhibitors of renal calcium reabsorption, and increased urinary calcium excretion. Additionally, an increase in LDL-C levels, fat accumulation, and changes in the expression of genes associated with chronic inflammation were observed, suggesting that HFCS may promote the development of LSRD. HFCS induced significant TAGE accumulation in the serum and liver, but not in the kidneys of our rat model. Furthermore, HFCS loading led to changes in gene expression in the kidneys without significant RAGE upregulation or TAGE accumulation. Interestingly, specific changes in mRNA expression in the kidneys correlated more with the TAGE levels in liver tissue than with those in the serum. Thus, direct or indirect interactions between the liver and kidneys—not involving serum TAGE or RAGE and signal transduction mediators, such as exosomes or cytokines, other than IL-1β and TNF-α—are suggested as potential factors contributing to the observed alterations in kidney gene expression.

## Figures and Tables

**Figure 1 nutrients-13-00080-f001:**
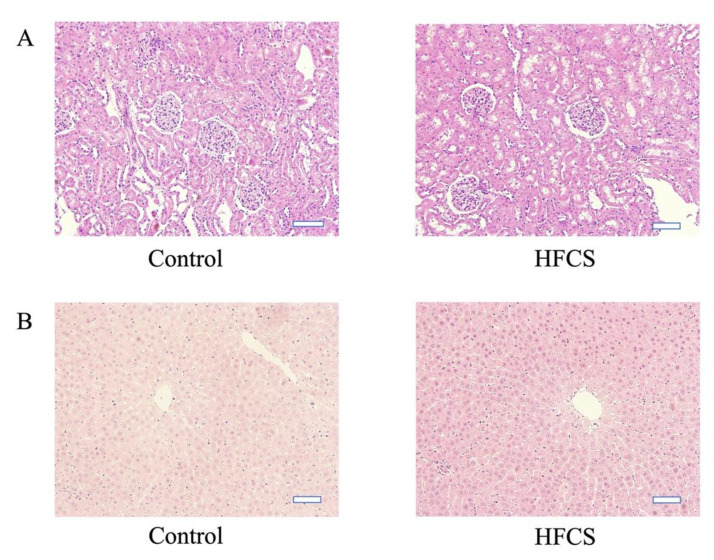
Representative images of hematoxylin and eosin-stained Wistar rat kidney (**A**) and liver (**B**) tissues from the high-fructose corn syrup (HFCS) and control groups. The scale bar represents 50 µm.

**Figure 2 nutrients-13-00080-f002:**
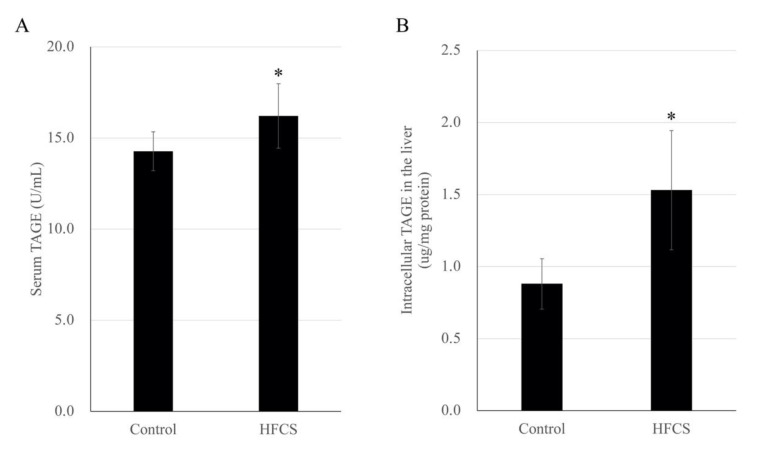
Quantitative analysis of serum toxic advanced glycation end product (TAGE) levels (**A**) and intracellular TAGE levels in the liver (**B**) in the high-fructose corn syrup (HFCS) and control groups of rats. Data are presented as the mean ± standard deviation (*n* = 10). * *p* < 0.05 HFCS versus control group (Mann-Whitney *U*-test).

**Figure 3 nutrients-13-00080-f003:**
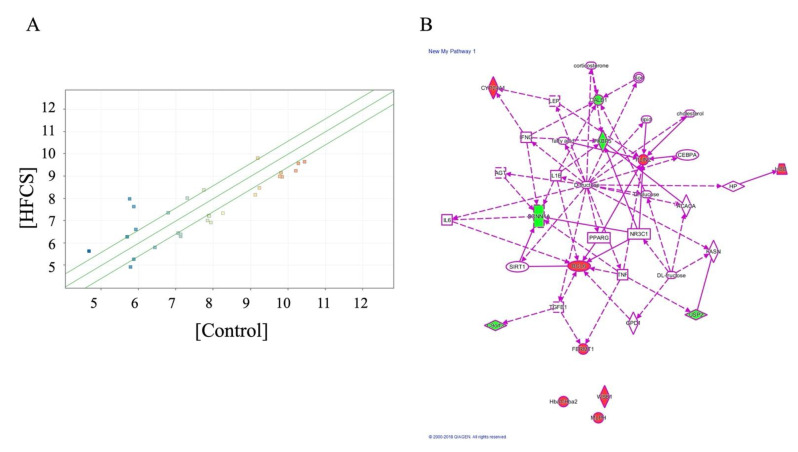
Analysis of high-fructose corn syrup (HFCS)-induced genes in the kidneys. (**A**) Volcano plot of HFCS-induced genes in the kidney, detected by DNA microarray analysis in the HFCS versus control group. The red dots indicate high gene expression levels, while the blue dots indicate low gene expression levels. (fold change >1.5; *p* < 0.05 by *t*-test). (**B**) Ingenuity pathway analysis-identified HFCS-induced gene network.

**Figure 4 nutrients-13-00080-f004:**
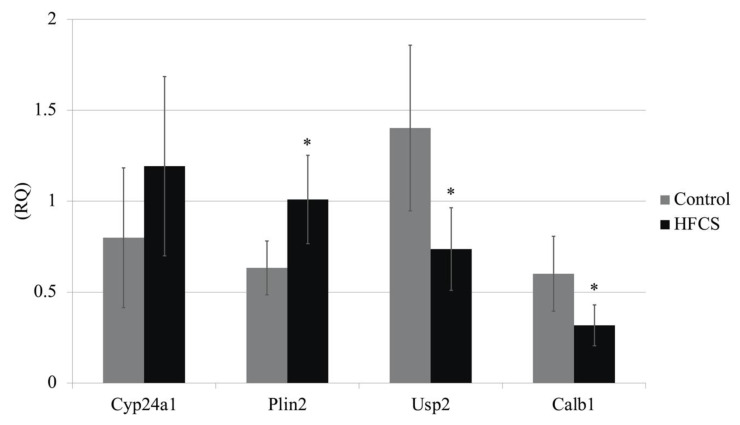
mRNA expression of *Cyp24a*, *Plin2*, *Usp2*, and *Calb1* genes in Wistar rat kidneys from the high-fructose corn syrup (HFCS) and control groups. Expression levels of the target genes were normalized to those of β-actin (*n* = 10 per group). * *p* < 0.05 HFCS versus control group (Mann-Whitney *U*-test). RQ, relative quantification by real-time polymerase chain reaction.

**Figure 5 nutrients-13-00080-f005:**
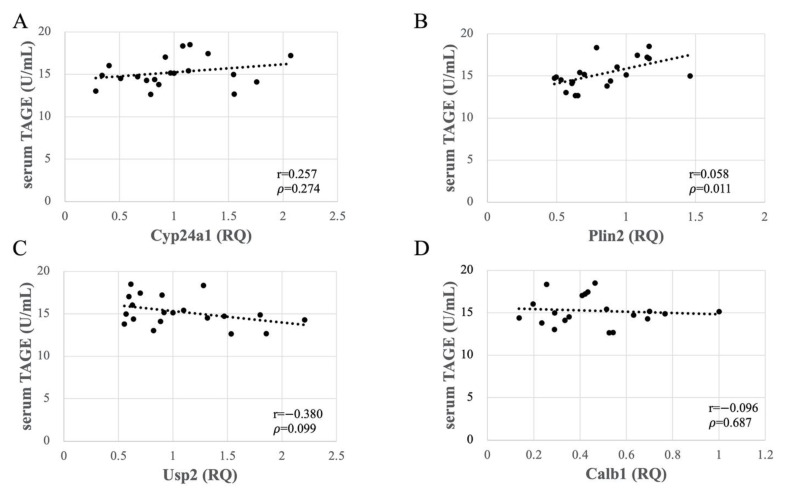
Relationships between serum toxic advanced glycation end-products (TAGE) concentrations and expression levels of *Cyp24a* (**A**), *Plin2* (**B**), *Usp2* (**C**), and *Calb1* (**D**) in Wistar rat kidneys. RQ, relative quantification by real-time polymerase chain reaction (*n* = 10 per group).

**Figure 6 nutrients-13-00080-f006:**
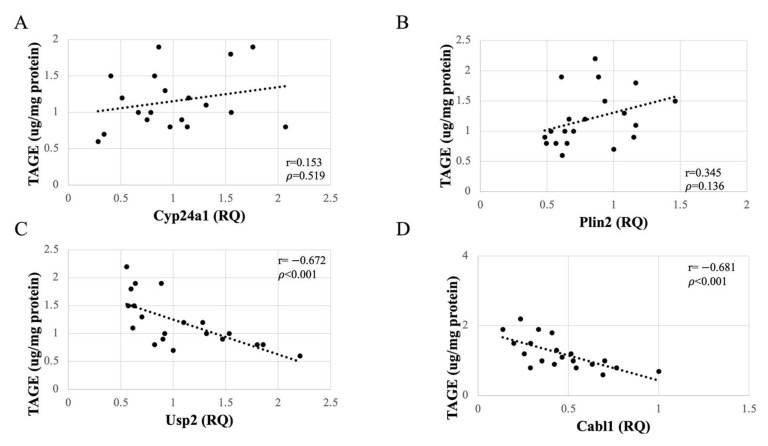
Relationships between intracellular toxic advanced glycation end-products (TAGE) levels in the liver of Wistar rats and expression of *Cyp24a* (**A**), *Plin2* (**B**), *Usp2* (**C**), and *Calb1* (**D**) in the kidneys. RQ, relative quantification by real-time polymerase chain reaction (*n* = 10 per group).

**Table 1 nutrients-13-00080-t001:** Upregulated genes in the high-fructose corn syrup (HFCS) versus control group (*n* = 3 each) based on microarray analysis.

Gene Symbol	Gene Description/Main Function of the Protein Encoded	HFCS/Control Fold Change
*Hbb*	Hemoglobin subunit beta/Involved in oxygen transport from the lung to the various peripheral tissues.	4.686
*Hba1/Hba2*	Hemoglobin, alpha 1/hemoglobin alpha, adult chain 2/The two nearly identical genes provide instructions for the synthesis of alpha-globin	1.983
*Cyp24a1*	Cytochrome P450, family 24, subfamily A, member 1/This enzyme helps control the amount of active vitamin D available in the body	1.669
*Bcl6*	B-cell lymphoma 6/Evolutionarily conserved zinc finger transcription factor containing an N-terminal POZ/BTB domain	1.620
*Plin2*	Perilipin 2/Adipose differentiation-related protein, also known as ADRP or adipophilin, belonging to the PAT family of cytoplasmic lipid droplet-binding proteins	1.576
*Wsb1*	WD repeat and SOCS box-containing 1/Substrate recognition protein within an E3 ubiquitin ligase, with the capability to bind diverse targets and mediate their ubiquitinylation and proteolytic degradation	1.549
*Mlph*	Melanophilin/Rab effector involved in melanosome transport.	1.509
*Fermt1*	Fermitin family member 1/Involved in integrin signaling and linkage of the actin cytoskeleton to the extracellular matrix	1.502

**Table 2 nutrients-13-00080-t002:** Downregulated genes in the high-fructose corn syrup (HFCS) versus control group (*n* = 3 each) based on microarray analysis.

Gene Symbol	Gene Description/Main Function of the Protein Encoded	HFCS/Control Fold Change
*Fkbp5*	FKBP prolyl isomerase 5/Member of the immunophilin protein family, playing a role in immunoregulation and basic cellular processes involving protein folding and trafficking	−1.758
*Calb1*	Calbindin 1/Member of the calcium-binding protein superfamily that includes calmodulin and troponin C	−1.714
*Usp2*	Ubiquitin-specific peptidase 2/Required for TNF-α-induced NF-κB signaling	−1.594
*Scnn1a*	Sodium channel epithelial 1 alpha subunit/Subunit of the epithelial sodium channel ENaC in vertebrates	−1.545

## Data Availability

The authors declare that the data supporting the findings of this study are available within the paper and its Appendix A.

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
