# Peer review of "Potential of an Interorgan Network Mediated by Toxic Advanced Glycation End-Products in a Rat Model"

_nutrients, 2020, doi:10.3390/nu13010080_

Round 1
Reviewer 1 Report
The present study by Shinya Inoue et al. examined the effect of high fructose corn syrup (HFCS) on kidney and liver dysfunction in Wistar rats specifically with elevated toxic advanced glycation end products (TAGE) in serum and tissues. They report changes in few genes involved in fructose metabolism, inflammation and lipid accumulation in the kidneys and attribute exosomes or cytokines the reason for these changes. The study has some good preliminary results but could be improved by following changes and experiments.1) For the benefit of readers, it would be good if authors could add some major functions of the genes involved in Table 1 and 2.
2) It would be good to confirm few of the genes by RT PCR in the liver tissue that the authors observed with microarray.
3) Authors looked at an inflammatory gene in the kidney, it would be more promising to run some more inflammatory markers by RT PCR and also macrophage infiltration in the kidneys by CD 68 staining.
4) The authors also looked at genes involved in lipid accumulation which is important component in this model but they should also look at lipid accumulation in the liver and kidney by ORO staining to confirm it.
5) Sentence formation, grammatical and spell check required.
Author Response
Response to Reviewer 1 Comments
Thank you very much for providing important comments. We are thankful for the time and energy you expended. Our responses to your comments are as follow:
We inserted New Figure; Figure S1, Figure S2, Figure S5 and Figure S6.
We changed the supplementary figure number (S1→S3, S2→S4).
We inserted new reference 21.
Therefore, the number of references was changed (e.g., 21→22, ) after new refence 22.
Point 1: For the benefit of readers, it would be good if authors could add some major functions of the genes involved in Table 1 and 2.
Response 1: Thanks for the advice. I have added a section "The main function of the protein or enzyme encoded by the gene" to Table 1 and Table 2.
Point 2: It would be good to confirm few of the genes by RT PCR in the liver tissue that the authors observed with microarray.
Response 2: This is a point that has been raised by other reviewers. We examined the expression of four genes in the liver by Realtime PCR. Ager, Plin2, and Usp2 were found to be expressed in the liver, but there was no significant difference in gene expression. In addition, we could not detect Cyp24a1 and Calb1 gene expression in our experimental system. These results suggest that HFCS intake causes completely different gene expression in the kidney and liver, and that this effect is specific to the kidney.
Gene expression of the liver: Gene expression of Ager, Plin2, Usp2 in Wister rat livers analysed with real-time polymerase chain reaction (RT-PCR). Normalised gene expression levels are shown as a ratio between the mean value for the target gene and that for β actin in each sample (n = 4 in each group). P values were based on Mann Whitney-U test. *P < 0.05 versus the control group. RQ, relative quantification.
Point 3: Authors looked at an inflammatory gene in the kidney, it would be more promising to run some more inflammatory markers by RT PCR and also macrophage infiltration in the kidneys by CD 68 staining.
Response 3: Thank you for your constructive feedback. This is a point that has been raised by other reviewers. To study inflammatory cytokines and gene expression in the kidney, we have performed CD68 immunostaining of kidneys, added to Materials and Methods section 2.10 (p.5, lines 200-206) and added to Results section 3.6 (p.10, lines306-308). Immunostaining images were also added as FigureS2. The immunostaining results showed that CD68 staining was negative in both the Control and HFCS groups.
However, we could not analysed some inflammatory markers in the kidney by RT-PCR, because the amount of the samples of renal tissues was too little.
Since TNF-α and IL-1β did not increased in the serum, we considered that dramatically inflammation might not occur in the organs such as kidney and liver.
Point 4: The authors also looked at genes involved in lipid accumulation, which is important component in this model, but they should also look at lipid accumulation in the liver and kidney by ORO staining to confirm it.
Response 4: Thank you for your opinion. We have performed Oil Red O staining on frozen kidney and liver sections and have added the results to Materials and Methods 2.8 (p4, lines185-192) and results 3.8 (p11, lines315-317). Immunostaining images were also added as FigureS1. Both liver and kidney were negative for ORO staining.
Point 5: Sentence formation, grammatical and spell check required.
Response 5: Thanks for the advice. We have rechecked the formatting of the text according to the instruction and had our manuscript checked by English native speaker before submission.

Reviewer 2 Report
The authors shed light on the possible interaction between liver and kidney in the context of the LSRD with a focus on TAGE. Although, the authors observed a correlation of hepatic TAGE with gene expression of usp2 and calb1, so the data situation is very small to speak of a direct or indirect interaction between the liver and kidney. With the obtained data the hypothesis and further the conclusion is not adequately met. Probably, the story should be reconsidered, if necessary the data would have to be reinterpreted or rearranged.
As the authors mentioned that usp2 is involved in inflammatory processes, it would also have been interesting to monitor inflammatory responses in order to correlate them if necessary. At least the measurement of cytokines in plasma is necessary to support the conclusion as authors mentioned it in the abstract, for example. In addition, the question arises why expression was only examined in the kidneys and not in the liver.
Moreover, the administration of HFCs does not result in any visible steatosis. Therefore, the results cannot be discussed in the context of steatosis or NAFLD. Similarly, there are no signs of inflammation shown at least in the histology, therefore to discuss results in context of NASH is also not adequate. Furthermore, the authors stated that HFCS ingestion caused fat accumulation. The reviewer can not follow this statement. If the authors meant increased TG plasma concentration, then it is not quite correct to say “fat accumulation” at least it was not occur in tissue.
Moreover, the protein expression of RAGE is more meaningful as the mRNA expression of it. Maybe there will be differences in the receptor expression when viewed at the protein level.
Author Response
Response to Reviewer 2 Comments
We wish to express our appreciation to you for your insightful comments on our paper. The comments have helped us significantly improve the paper.
We inserted New Figure; Figure S1, Figure S2, Figure S5 and Figure S6.
We changed the supplementary figure number (S1→S3, S2→S4).
We inserted new reference 21.
Therefore, the number of reference was changed (e.g. 21→22, ) after new refence 22.
Point 1: The authors shed light on the possible interaction between liver and kidney in the context of the LSRD with a focus on TAGE. Although, the authors observed a correlation of hepatic TAGE with gene expression of usp2 and calb1, so the data situation is very small to speak of a direct or indirect interaction between the liver and kidney. With the obtained data the hypothesis and further the conclusion is not adequately met. Probably, the story should be reconsidered, if necessary the data would have to be reinterpreted or rearranged.
Response 1: Thank you for providing these insights. We considered the correlation between TAGE in the liver and Usp2 and Calb1 gene expression in the kidney as follows.
Since the intracellular TAGE in the kidney was below the detection limit, it is unlikely that the TAGE production affected the expression of USP2 and Calb1 genes. Serum TAGE increased 1.14-fold in the HFCS group compared to the Control group. On the other hand, there was no difference in RAGE gene expression in the kidney. Therefore, we hypothesized that blood TAGE decreased the gene expression of USP2 and Calb1. However, there was no correlation between blood TAGE and USP2 and Calb1 gene expression.
Otherwise, we found a significant negative correlation between TAGE in liver and USP2 and Calb1 gene expression. It is possible that the phenomenon caused by TAGE production in the liver induced the decrease in USP2 and Calb1 gene expression in the kidney.
To make this point clearer, we have added the following to the Discussion
Point 2: As the authors mentioned that usp2 is involved in inflammatory processes, it would also have been interesting to monitor inflammatory responses in order to correlate them if necessary. At least the measurement of cytokines in plasma is necessary to support the conclusion as authors mentioned it in the abstract, for example. In addition, the question arises why expression was only examined in the kidneys and not in the liver.
Response 2: I greatly appreciate your constructive comments. Your point is well taken. This is a point that has been raised by other reviewers. To study inflammatory cytokines and gene expression in the kidney, we performed CD68 immunostaining in the kidney and serum IL-1b and TNF-α measurements. Regarding CD68 immunostaining, we added it to 2.10 in Materials and methods (p.5, line 200-206) and 3.6. in the results section (p.10, lines 306-308) , and serum cytokines have been added to 2.9 of Materials and Methods (p.5, lines 194-199) and 3.6 of the results section (p.11, lines 310-314) . Immunostaining images were also added as FigureS2. The immunostaining results showed that CD68 staining was negative in both the Control and HFCS groups. The results of IL-1β were shown in Figure S6. IL-1β tended to be higher in the HFCS group, but the difference was not statistically significant. TGF-α was below the detection sensitivity in both groups.
Gene expression in the liver was also examined, as pointed out by other reviewers regarding gene statements in the liver. We examined the expression of four genes in the liver by Realtime PCR. Ager, Plin2, and Usp2 were found to be expressed in the liver, but there was no significant difference in gene expression. In addition, we could not detect Cyp24a1 and Calb1 gene expression in our experimental system. These results suggest that HFCS intake causes completely different gene expression in the kidney and liver, and that this effect is specific to the kidney.
We removed “cytokines” in the abstract and conclusion.
We inserted “cytokines without IL-1β and TNF-α“ in the abstract and conclusion.
Gene expression of the liver: Gene expression of Ager, Plin2, Usp2 in Wister rat livers analysed with real-time polymerase chain reaction (RT-PCR). Normalised gene expression levels are shown as a ratio between the mean value for the target gene and that for β actin in each sample (n = 4 in each group). P values were based on Mann Whitney-U test. *P < 0.05 versus the control group. RQ, relative quantification.
Point 3: Moreover, the administration of HFCs does not result in any visible steatosis. Therefore, the results cannot be discussed in the context of steatosis or NAFLD. Similarly, there are no signs of inflammation shown at least in the histology, therefore to discuss results in context of NASH is also not adequate. Furthermore, the authors stated that HFCS ingestion caused fat accumulation. The reviewer cannot follow this statement. If the authors meant increased TG plasma concentration, then it is not quite correct to say “fat accumulation” at least it was not occurred in tissue.
Response 3: Thank you for your meaningful reply. Based on reviewer comments, we removed the sentences for “ fat accumulation” in the previous manuscripts. In the liver specimens of our animal model, no histological changes were observed in the Control and HFCS groups. Regarding Oil Red O staining, we added it to 2.10 in Materials and methods (p.5, line 200-206) and 3.8. in the results section (p.4-5, lines 185-192) . Typical images of Oil Red O stain in the kidney and the liver were also added as Figure S6. In addition, Oil Red O staining did not show any fat droplets in either group. These results support that no inflammatory changes are occurring at this time. This is as you pointed out. We described relationship Plin2, Usp2 and TG. We have rewritten to be more in line with your comments (p.11, 12).
Point 4: Moreover, the protein expression of RAGE is more meaningful as the mRNA expression of it. Maybe there will be differences in the receptor expression when viewed at the protein level.
Response 4: I followed your advice and did a Western blot on RAGE. We added it to 2.6 and 2.7 in Materials and methods (p.4, lines222-247) and 3.3. in the results section (p.7, lines 320-324.) Western blot results are shown in Figure S5. The Western blot results were also consistent with the PT-PCR results, showing no significant difference in RAGE protein expression in the kidney.

Reviewer 3 Report
Manuscript „Identification of interorgan netwprk meditaed by toxic advanced glycation end-products in a rat model by Inoue et al., has been performed on in vivo rat model and provides data focused predominantly on intracellular effects of toxic advanced glycation-end products (TAGE) in kidney. Since dysfunctions caused by HFCS consumption are broadly studied in liver, than experimental focus on kidney as the object for the research on intracellular effects exerted by excessive sugar intake enabled, to some extent, to fill the gap concerning the specific metabolic aspects of kidney activity. In this study, 8-weeks lasting consumption of high-fructose corn syrup resulted in TAGE accumulation both in the serum and liver of male rats as well as in a significant changes of 12 annotated kidney genes expression level ( Hbb, Hba1/ Hba2, Cyp24a1, Bcl6, Plin2,Wsb1, Mph, Fermf1 were up- regulted whereas Fkbp5, Calb1, Usp2 and Scnn1a were down-regulated). By affecting Cyp24a1 and Calb1 genes activity, HFCS ingestion altered genes encoding inhibitors of renal calcium reabsorption an induced an increase in urinary calcium excretion. Authors also report that in kidneys changes in selected genes mRNA expression levels occurred without TAGE accumulation or receptor for advanced -glycation end-products (RAGE) up-regulation and were more correlated with TAGE levels in liver than in serum. An elevated level of LDL, cholesterol as well as fat accumulation in HFCS receiving animals was associated with chronic inflammation of their fat tissue.
Although paper provides valuable and interesting data, is well written, the text of the manuscript would benefit from some minor introduced changes and explanations
- What was the rationale to choose 8 weeks long lasting. period of HFCS supplementation? Could it be possible that the lack of some HFCS-induced effects found at the kidney level might be dependent on the length of period with high glucose ingestion?
- Authors conclude that exosomes might have affected kidney gene expression. Since their research did not provide any data concerning this aspect, a more detailed explanations concerning a possible mechanism through which exosomes affect gene expression would be appreciated (see lines 310-314). In its present form, text in lines 310-314 appears as a low informative.
3.If changes were statistically insignificant, then an expression „insignificant differences” used in line 163 is incorrect.
Author Response
Response to Reviewer 3 Comments
We thank you for careful reading our manuscript and for your useful/helpful suggestions.
We inserted New Figure; Figure S1, Figure S2, Figure S5 and Figure S6.
We changed the supplementary figure number (S1→S3, S2→S4).
We inserted new reference 21.
Therefore, the number of reference was changed (e.g. 21→22, ) after new refence 22.
Point 1: What was the rationale to choose 8 weeks long lasting. period of HFCS supplementation? Could it be possible that the lack of some HFCS-induced effects found at the kidney level might be dependent on the length of period with high glucose ingestion?
Response 1: Thanks for the question.
There are some reports for the WBN/Kob-Lepfa rats and WBN/kob rats. Kaji et al. reported non-fasted plasma glucose and TG levels in male WBN/Kob rats (11 weeks) and age-matched male WBN/Kob-Leprfa rats, which are a type 2 diabetes mellitus (T2DM) model in the pre-diabetic phase [1]. Plasma glucose levels in WBN/Kob rats and WBN/Kob-Leprfa rats were approximately 6.7 and 10 mM, respectively. Although plasma glucose levels were approximately 150% higher in WBN/Kob-Leprfa rats than in WBN/Kob rats, they did not develop T2DM. Furthermore, non-fasted plasma TG levels in WBN/Kob rats and WBN/Kob-Leprfa rats were approximately 1.3 and 5.8 mM, respectively. There was no description regarding whether WBN/Kob-Leprfa rats developed dyslipidemia. In another study, male Wister rats (7 weeks) and age-matched male WBN/Kob-Leprfa rats were bred and fed standard rat chow for four weeks, and their fasted plasma TG levels were approximately 0.6 and 4.5 mM, respectively [2]. So we the rearing period of the rats as 8 weeks.
Takata et al. reported that Witer/ST rats (11 weeks) drunk 10% HFCS for 13 weeks.
Then serum TAGE level and TAGE in the liver of HFCS group increased approximately 1.5 and 3.0-fold compared with control group.
Since the serum TAGE level is the high sensitivity marker, we think that we could observe effects of the rats which drunk HFCS for 8weeks.
[1] : Kaji, N.; Okuno, A.; Ohno-Ichiki, K.; Oki, H.; Ishizawa, H.; Shirai, M.; Asai, F. Plasma profiles of glucose, insulin, and lipids in the male WBN/Kob-Leprfa rat, a new model of type 2 diabetes with obesity. J. Vet. Med. Sci. 2012, 74, 1185–1189.
[2] : Namekawa, J.; Takagi, Y.; Wakabayashi, K.; Nakamura, Y.; Watanabe, A.; Nagakubo, D.; Shirai, M.; Asai, F. Effects of high-fat diet and fructose-rich diet on obesity, dyslipidemia and hyperglycemia in the WBN/Kob-Leprfa rat, a new model of type 2 diabetes mellitus. J. Vet. Med. Sci. 2017, 79, 988–991.
Point 2: Authors conclude that exosomes might have affected kidney gene expression. Since their research did not provide any data concerning this aspect, a more detailed explanations concerning a possible mechanism through which exosomes affect gene expression would be appreciated (see lines 310-314). In its present form, text in lines 310-314 appears as a low informative.
Response 2:
We tried to analyze the dysfunction or risk factors of kidney by some methods.
Methods 1: We analyzed the intracellular TAGE in the kidney. However intracellular TAGE are not able to detect.
Method 2: We analyzed serum levels of TAGE and expression of RAGE in the kidney because we considered the possibility. TAGE-RAGE system might induced gene expression of kidney.
However, we observed no correlation of serum TAGE and gene expression of kidney.
Methods 3: We analyzed intracelluer TAGE in the liver and they correlated gene expression of kidney. Therefore we considered factors associated with liver which generatededs intracelluer TAGE might have induced gene expression of kidney.
Exosome and cytokines may belong to this factor.
This is a new manuscript detailing these steps.
Point 3: If changes were statistically insignificant, then an expression ‘insignificant differences” used in line 163 is incorrect.
Response 3: We rewrote the sentence from "insiginificant differences" to "No significant differences".

Round 2
Reviewer 1 Report
Major comments addressed by the authors. The manuscript looks good compared to original submission.
Author Response
Response to Reviewer 1 Comments
We would like to thank both of the Reviewers for their comments on our manuscript. We addressed the comments provided and hope that our responses are satisfactory. Point-by-point responses are listed below.
In our previous manuscript, the title was “Identification of an interorgan network mediated by toxic advanced glycation end-products in a rat model”. However, the revised manuscript is entitled “Potential of an interorgan network mediated by toxic advanced glycation end-products in a rat model”.
We changed a figure of the positive control in supplementary figure S1.
We inserted new reference 55 and 59.
Therefore, the number of reference was changed (e.g. 55→56, ) after new refence 55.
Before submitting this paper, it was evaluated by a professional English proofreader. No changes have been made to the content, but the entire text has been corrected for spelling, grammar, and other correct English expressions.
New sentences/words are highlighted in yellow.
Point 1: English language and style are fine/minor spell check required
Response 1: I greatly appreciate your constructive comments. The revised manuscript and Figures were proofread by a native speaker of English.
Reviewer 2 Report
Again, the administration of HFCs does not result in any visible steatosis. Therefore, the results cannot be discussed in the context of steatosis or NAFLD. Again, there are no signs of inflammation shown at least in the histology, or now in the cytokine analyzes, therefore to discuss results in context of NASH is not adequate. Indeed, the authors tried to address this issue and tried to change it in the section discussion but still the problem exists. I think the story of the manuscript have to be changed. This is not a model of NAFLD or NASH.
The Oil redO staining quality in particular the positive control is weak.
As the authors wrote above: "The phenomenon caused by TAGE production in the liver induced the decrease in USP2 and Calb1 gene expression in the kidney." Still, data situation is too weak to speak of a direct interaction between the liver and kidney. There are no evidences, in my opinion just a correlation is not enough.
Author Response
Response to Reviewer 2 Comments
We would like to thank both of the Reviewers for their comments on our manuscript. We addressed the comments provided and hope that our responses are satisfactory. Point-by-point responses are listed below.
In our previous manuscript, the title was “Identification of an interorgan network mediated by toxic advanced glycation end-products in a rat model”. However, the revised manuscript is entitled “Potential of an interorgan network mediated by toxic advanced glycation end-products in a rat model”.
We changed a figure of the positive control in supplementary figure S1.
We inserted new reference 55 and 59.
Therefore, the number of reference was changed (e.g. 55→56, ) after new refence 55.
Before submitting this paper, it was evaluated by a professional English proofreader. No changes have been made to the content, but the entire text has been corrected for spelling, grammar, and other correct English expressions.
New sentences/words are highlighted in yellow.
Point 1: Again, the administration of HFCs does not result in any visible steatosis. Therefore, the results cannot be discussed in the context of steatosis or NAFLD. Again, there are no signs of inflammation shown at least in the histology, or now in the cytokine analyzes, therefore to discuss results in context of NASH is not adequate. Indeed, the authors tried to address this issue and tried to change it in the section discussion but still the problem exists. I think the story of the manuscript have to be changed. This is not a model of NAFLD or NASH.
Response 1: Thank you for providing these insights. As you pointed out, there were no findings suggesting steatosis or NAFLD due to HFCS intake in this study. In discussing the results, we tried to improve the argument that we had made a leap to the condition such as NAFLD, which did not occur. We have therefore deleted lines 341-343 on page 12 and lines 352-370 on page 13, and added a new discussion of TAGE in lines 348-352 and 370-376 on page 13.
We have deleted lines 405-406, 408-410, and 424-428 on page 14 regarding genes induced by HFCS ingestion. In accordance with this, a discussion with the addition of reference 55 was added on page 14, lines 417-422.
Point 2: The Oil redO staining quality in particular the positive control is weak.
Response 2: I greatly appreciate your constructive comments. We changed a figure of the positive control in supplementary figure S1.
Point 3: As the authors wrote above: "The phenomenon caused by TAGE production in the liver induced the decrease in USP2 and Calb1 gene expression in the kidney." Still, data situation is too weak to speak of a direct interaction between the liver and kidney. There are no evidences, in my opinion just a correlation is not enough.
Response 3: Thank you for your meaningful reply. As you pointed out, the data situation is too weak to talk about a direct interaction between liver and kidney. First of all, we have changed the title of this paper. We have also deleted lines 433-435 on page 14. We have added 59 new references necessary for the development of the argument and a new discussion on page 14, lines 441-444.
